# Werner syndrome helicase is a selective vulnerability of microsatellite instability-high tumor cells

Simone Lieb[1†], Silvia Blaha-Ostermann[1†], Elisabeth Kamper[1], Janine Rippka[1], Cornelia Schwarz[1], Katharina Ehrenhöfer-Wölfer[1], Andreas Schlattl[1], Andreas Wernitznig[1], Jesse J Lipp[1], Kota Nagasaka[2], Petra van der Lelij[2], Gerd Bader[1], Minoru Koi[3], Ajay Goel[4], Ralph A Neumüller[1], Jan-Michael Peters[2], Norbert Kraut[1], Mark A Pearson[1], Mark Petronczki[1]*, Simon Wöhrle[1]*

[1]Boehringer Ingelheim RCV GmbH & Co KG, Vienna, Austria; [2]Research Institute of Molecular Pathology, Vienna, Austria; [3]Division of Gastroenterology, Department of Internal Medicine, Comprehensive Cancer Center, University of Michigan, Ann Arbor, United States; [4]Center for Gastrointestinal Research, Baylor Scott & White Research Institute and Charles A. Sammons Cancer Center, Baylor University Medical Center, Dallas, United States

*For correspondence:
mark_paul.petronczki@
boehringer-ingelheim.com (MP);
simon.woehrle@boehringer-
ingelheim.com (SW)

†These authors contributed
equally to this work

Competing interest: See
page 18

Reviewing editor: Wolf-Dietrich
Heyer, University of California,
Davis, United States

**Abstract** Targeted cancer therapy is based on exploiting selective dependencies of tumor cells. By leveraging recent functional screening data of cancer cell lines we identify Werner syndrome helicase (WRN) as a novel specific vulnerability of microsatellite instability-high (MSI-H) cancer cells. MSI, caused by defective mismatch repair (MMR), occurs frequently in colorectal, endometrial and gastric cancers. We demonstrate that WRN inactivation selectively impairs the viability of MSI-H but not microsatellite stable (MSS) colorectal and endometrial cancer cell lines. In MSI-H cells, WRN loss results in severe genome integrity defects. ATP-binding deficient variants of WRN fail to rescue the viability phenotype of WRN-depleted MSI-H cancer cells. Reconstitution and depletion studies indicate that WRN dependence is not attributable to acute loss of MMR gene function but might arise during sustained MMR-deficiency. Our study suggests that pharmacological inhibition of WRN helicase function represents an opportunity to develop a novel targeted therapy for MSI-H cancers.
DOI: https://doi.org/10.7554/eLife.43333.001

## Introduction

Defects in components of the DNA repair machinery, such as *BRCA1/2* mutations or impaired DNA mismatch repair (MMR), are a common characteristic of tumor cells, accelerating the accumulation of DNA mutations or chromosomal aberrations that are required for neoplastic growth and transformation (*Kinzler and Vogelstein, 1997*). Plasticity of genome stability pathways permits tumor cells to tolerate the loss of individual DNA repair genes and leads to synthetic lethality (SL) upon targeting the compensating repair mechanism (*Nickoloff et al., 2017*). The first clinically approved drugs exploiting such a SL interaction are Poly(ADP-Ribose) Polymerase (PARP) inhibitors for therapy of BRCA1/BRCA2-deficient tumors (*Kaufman et al., 2015*; *Lord and Ashworth, 2017*).

MMR deficiency is caused by inactivation of genes of the DNA repair machinery involved in the resolution of nucleotide base-base mismatches during DNA replication (*Jiricny, 2006*; *Kunkel and Erie, 2015*). MMR defects lead to characteristic variations in the length of tandem nucleotide repeats across the genome, known as microsatellite instability (MSI) (*Ellegren, 2004*). Germline mutations in MMR genes, most commonly MLH1, MSH2, MSH6 and PMS2, are causative for Lynch

syndrome, a cancer predisposition condition associated with increased lifetime risk to develop colo-rectal cancer (CRC) or other tumor types including endometrial and gastric carcinoma (*Hampel et al., 2005*; *Lynch and Krush, 1971*; *Lynch et al., 2015*). In sporadic, nonhereditary CRC, MSI is frequently observed due to epigenetic silencing of MLH1 (*Cunningham et al., 1998*; *Herman et al., 1998*; *Kane et al., 1997*; *Kuismanen et al., 2000*). MSI-high (MSI-H) tumors display a hypermutator phenotype (*Cancer Genome Atlas Network, 2012*), which entails increased immu-nogenicity, amendable to therapy with immune checkpoint inhibitors (*Le et al., 2015*). However, tar-geted therapies directly exploiting the MMR-deficient status of tumor cells do not exist.

Werner syndrome helicase (WRN) is a member of the RecQ DNA helicase subfamily (*Croteau et al., 2014*; *Yu et al., 1996*). RecQ helicases are involved in multiple DNA processing steps including DNA replication, double-strand break repair, transcription and telomere mainte-nance and are therefore considered to serve as 'genome caretakers' (*Chu and Hickson, 2009*; *Croteau et al., 2014*). The critical function of this protein family in genome maintenance is under-scored by the fact that defects in three of the five family members – WRN, Bloom Syndrome RecQ Like Helicase (BLM) and RecQ Like Helicase 4 (RECQL4) – give rise to human disease syndromes associated with developmental defects and cancer predisposition (*Brosh, 2013*; *Oshima et al., 2017*). Specifically, patients with Werner syndrome display a premature ageing phenotype including arteriosclerosis, type II diabetes and osteoporosis and are prone to develop tumors of mesenchymal origin, such as soft tissue sarcoma or osteosarcoma (*Goto et al., 2013*; *Hickson, 2003*; *Lauper et al., 2013*). WRN is unique among RecQ family helicases in possessing $3'-5'$ exonuclease activity (*Huang et al., 1998*; *Kamath-Loeb et al., 1998*; *Shen et al., 1998*).

In contrast to the previously described tumor-suppressive role of WRN, we demonstrate in this study that WRN possesses a context-dependent critical pro-survival function for cancer cells. By leveraging a recently defined map of cancer cell specific vulnerabilities (*McDonald et al., 2017*) and a comprehensive molecular characterization of cancer cell models (*Barretina et al., 2012*; *Streit et al., 2019*) we identify WRN helicase as a selective dependency in MSI-H cancer cell lines.

## Results

### WRN dependency is associated with MSI-H status of cancer cells

WRN was identified as a potential selective dependency in a subset of 398 cancer cell models in a recent pooled shRNA viability screen covering approximately 8000 genes (Project DRIVE) (https://oncologynibr.shinyapps.io/drive/; *McDonald et al., 2017*). A genomic or expression-based bio-marker predictive for WRN dependency was unknown. Depletion of WRN exclusively affects viability of a subset of CRC, gastric and endometrial cancer cell models reflected by RSA (<u>r</u>edundant <u>si</u>RNA <u>a</u>ctivity) sensitivity scores $\leq -3$, indicative of cell essentiality (*Figure 1A*). Intriguingly, CRC, gastric and endometrial cancers are the three human malignancies with the highest frequency of MSI-H sta-tus (*Cortes-Ciriano et al., 2017*). This raised the possibility that WRN represents a selective depen-dency in MSI-H cell lines.

In order to explore this hypothesis we developed a Random Forest model using an MSI feature list defined by Boland and Goel (*Boland and Goel, 2010*). This model classifies WRN sensitive and insensitive cell lines with an accuracy of 0.89 and a recall rate for sensitive lines of 0.69. Importantly, no true insensitive cell lines are classified as sensitive (*Figure 1—figure supplement 1A*). An analysis of variable importance revealed MLH1 expression as the feature most highly associated with the classification outcome, in line with the frequent inactivation of the MLH1 gene in MSI-H CRC (*Cunningham et al., 1998*; *Herman et al., 1998*; *Kane et al., 1997*; *Kuismanen et al., 2000*) (*Fig-ure 1—figure supplement 1A*). Consistently, WRN dependency anti-correlates with MLH1 mRNA expression levels among the cell models used in Project DRIVE (*Figure 1—figure supplement 1B*; $p=1.02*10^{-4}$, Fisher's exact test, stratification of MLH1-low and -high expressing cell models accord-ing to median MLH1 expression [TPM 37.44]).

Next, we wanted to experimentally validate the MSI status in a select set of cell lines. To this end, we used a fluorescent PCR-based analysis of five mononucleotide microsatellite markers to deter-mine the MSS/MSI-H status of a subset of CRC, gastric and endometrial cancer cell models (*Supplementary file 1*). In addition, we utilized a comprehensive MSS/MSI-H status annotation of CRC cell models reported by Medico and colleagues (*Medico et al., 2015*). Analysis of gene

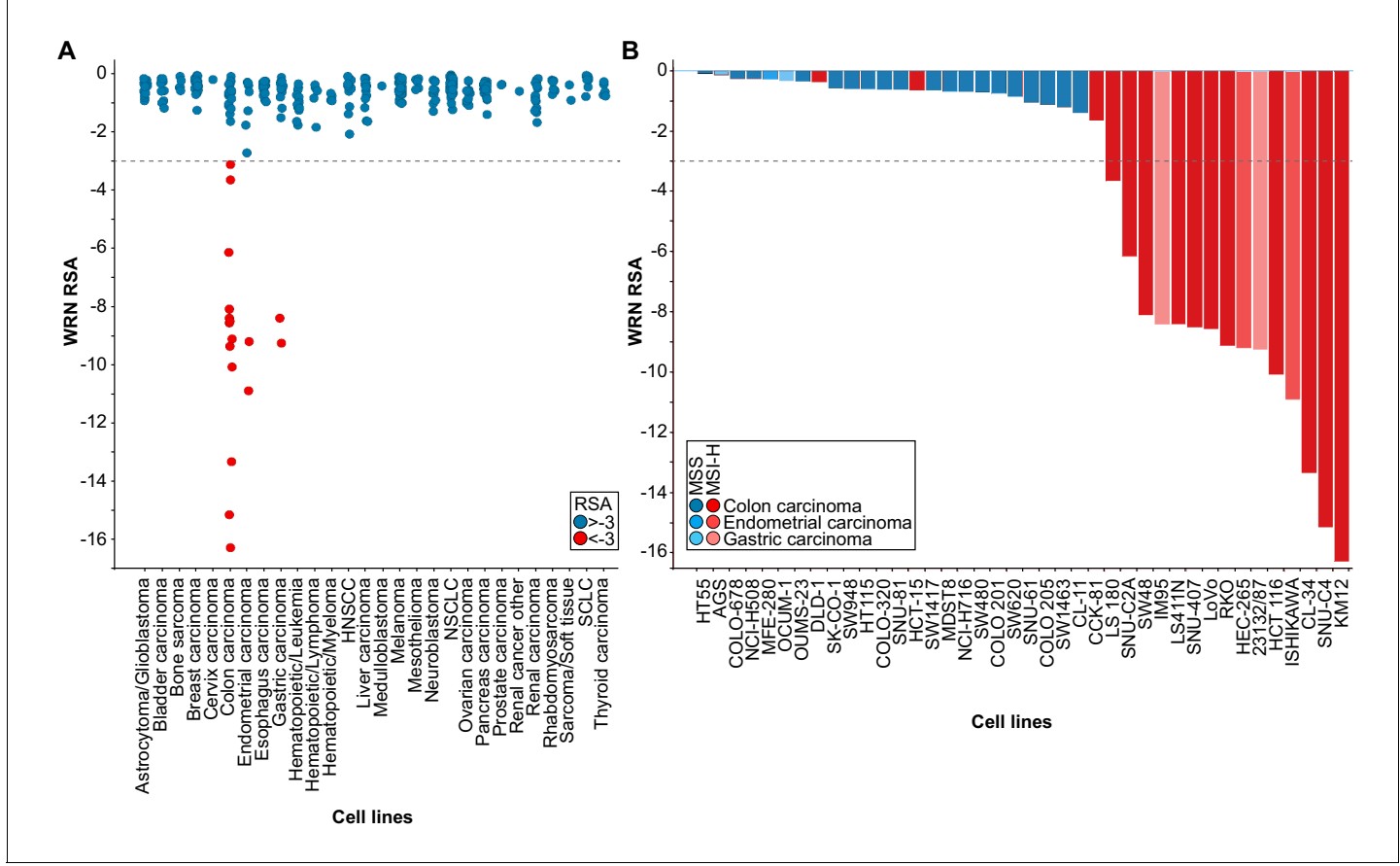

**Figure 1.** WRN is a selective dependency in MSI-H cancer cell models. (**A**) WRN shRNA activity by RSA score in pooled shRNA depletion screens from Project DRIVE (*McDonald et al., 2017*). Cell lines were binned according to tumor type. (**B**) MSS/MSI-H status and WRN RSA of CRC, endometrial and gastric cancer models from Project DRIVE.

DOI: https://doi.org/10.7554/eLife.43333.002

The following figure supplement is available for figure 1:

**Figure supplement 1.** WRN dependency correlates with MMR gene mutation status and MLH1 expression.
DOI: https://doi.org/10.7554/eLife.43333.003

dependency and MSS/MSI-H status data revealed that WRN dependency was strongly associated with MSI-H status across CRC, gastric and endometrial models (p=4.91*10$^{-8}$, Fisher's exact test). Of the 18 cell lines classified as MSI-H, 15 cell lines (83%) were sensitive to WRN depletion using an RSA value of $\leq -3$ to define WRN dependency (*Figure 1B*). In contrast, WRN is dispensable for viability in all MSS cell models (*Figure 1B*). Our analysis suggests that MSI-H status is a strong predictor for WRN sensitivity of cancer cells.

## WRN depletion by siRNA selectively impairs viability of MSI-H CRC and endometrial cancer cell lines

To experimentally corroborate the WRN dependency of MSI-H cancer cells, we applied short-interfering RNA (siRNA)-mediated knock-down of WRN in a panel of three MSS (SK-CO-1, CaCo-2, SW480) and three MSI-H (HCT 116, RKO, SNU-C4) CRC cell lines. In agreement with the results from Project DRIVE, WRN depletion using a mixture of four siRNA duplexes (Pool) or an individual siRNA (#1) targeting WRN profoundly affected viability in MSI-H, but not in MSS CRC models (*Figure 2A*). In contrast, depletion of the known essential mitotic kinase PLK1, had a detrimental effect on viability of both MSS and MSI-H cell lines. Efficient depletion of WRN protein following siRNA transfection was confirmed by immunoblotting (*Figure 2A*). The selective dependency on WRN was mirrored in colony formation assays with two MSS (LS1034, SK-CO-1) and two MSI-H cell lines (HCT 116, RKO)

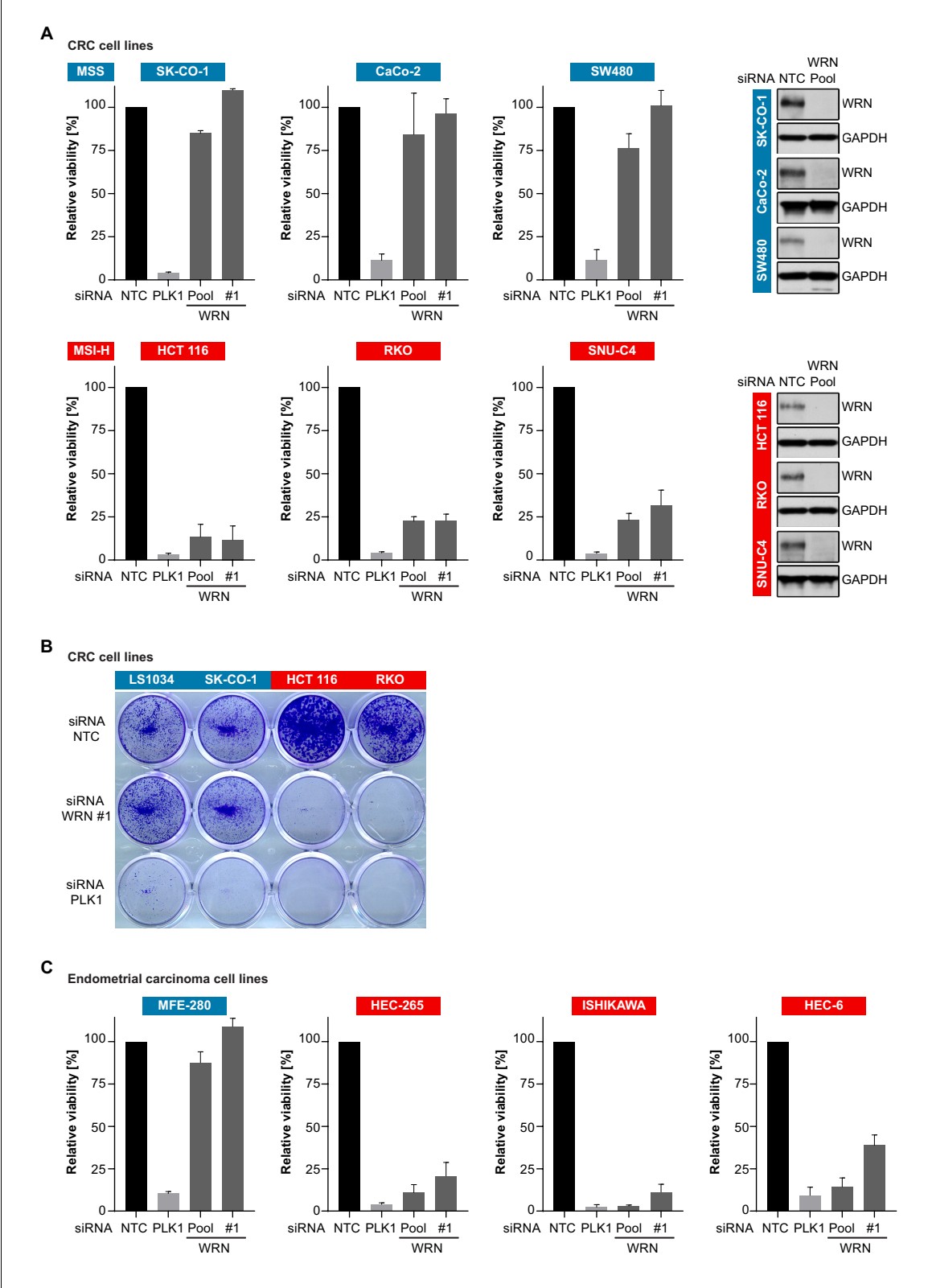

**Figure 2.** Loss of WRN selectively impairs viability of MSI-H CRC and endometrial cancer cell models. (**A**) MSS and MSI-H CRC cell lines were transfected with the indicated siRNAs. Cell viability was determined 7 days after transfection and is shown relative to non-targeting control (NTC) siRNA. WRN siRNA knock-down efficacy was analyzed by immunoblotting. Protein lysates were prepared 72 hr after transfection. GAPDH expression
*Figure 2 continued on next page*

*Figure 2 continued*

was used to monitor equal loading. (**B**) Crystal violet staining of MSS and MSI-H CRC lines treated as in panel A. (**C**) Cell viability analysis of MSS and MSI-H endometrial cell lines treated as in panel A. Data information: In (**A** and **C**), data are presented as mean ± SD of three independent experiments.

DOI: https://doi.org/10.7554/eLife.43333.004

The following figure supplements are available for figure 2:

**Figure supplement 1.** Non-transformed cells do not display WRN dependency.

DOI: https://doi.org/10.7554/eLife.43333.005

**Figure supplement 2.** MLH1/MSH3 reconstitution in HCT 116 CRC cells does not alleviate WRN dependency.

DOI: https://doi.org/10.7554/eLife.43333.006

**Figure supplement 3.** WRN inactivation does not elicit dependency on MLH1 or MSH3 in MSS SW480 CRC cells.

DOI: https://doi.org/10.7554/eLife.43333.007

(*Figure 2B*). Likewise, we observed that WRN knock-down impaired viability of three MSI-H endometrial carcinoma cells lines (HEC-265, ISHIKAWA, HEC-6), but not the MSS cell line MFE-280 (*Figure 2C*). WRN mRNA levels were similarly reduced upon transfection of pooled or individual WRN-targeting siRNAs in all four endometrial carcinoma models (*Figure 2—figure supplement 1A*).

Similar to MSS cancer models, non-transformed telomerase-immortalized human retinal pigment epithelial cells (hTERT RPE-1) did not display sensitivity to knock-down of WRN (*Figure 2—figure supplement 1B*). It is noteworthy, that the depletion of the related RecQ helicase BLM significantly impaired viability of hTERT RPE-1, but not HCT 116 cells. These RNAi experiments demonstrate that depletion of WRN abrogates viability in MSI-H but not MSS or non-transformed cells. To assess a potential mechanism to bypass WRN dependence in MSI-H cancer cells, we tested whether co-depletion of p53 and WRN in the *TP53*-wild-type MSI-H CRC line HCT 116 would reverse the sensitivity to WRN knock-down. However, WRN/p53 co-depletion exacerbated the reduction in viability compared to WRN knock-down alone (*Figure 2—figure supplement 1C*). *TP53*-wild-type MSS CRC SK-CO-1 cells were affected neither by individual or dual knock-down of WRN and p53. Interestingly, in both cell lines we observed a slight elevation of p53 protein levels upon WRN depletion (*Figure 2—figure supplement 1C*).

To determine whether reconstitution of MLH1 and MSH3 in the MSI-H CRC line HCT 116 would revert WRN dependence, we utilized variants of HCT 116 harboring copies of human chromosome 2 (+ch2, control line), chromosome 3 (+ch3, MLH1 reconstitution) or chromosome 3 and 5 (+ch3+ch5, MLH1 and MSH3 reconstitution) derived from normal fibroblasts (*Haugen et al., 2008*; *Koi et al., 1994*). In both HCT 116 +ch3 and HCT 116 +ch3+ch5 cell lines, we noticed only a modest increase of viability upon WRN knock-down compared to the parental cells or the HCT 116 +ch2 control line (*Figure 2—figure supplement 2A*) despite equally potent WRN knock-down in all models (*Figure 2—figure supplement 2B*). Reconstitution of MLH1 and MSH3 expression in HCT 116 +ch3 and HCT 116 +ch3+ch5 was confirmed by immunoblotting (*Figure 2—figure supplement 2C*). In a reciprocal approach, we monitored whether individual or combined knock-down of MLH1 and MSH3 would elicit differential effects on the viability of WRN-proficient versus WRN-knock-out cell models. To this end, using CRISPR-Cas9 we generated two WRN knock-out SW480 monoclonal lines (WRN KO #1 and #2) and a non-edited, WRN-proficient SW480 monoclonal control line (parental). Loss of WRN protein in the two knock-out lines was confirmed using immunoblot assays (*Figure 2—figure supplement 3A*). We did not observe an effect of either single or concomitant knockdown of MLH1 and MSH3 on cell viability of the WRN-deficient SW480 cell lines (*Figure 2—figure supplement 3A*), despite efficacious depletion of the mRNAs encoding MLH1 and MSH3 (*Figure 2—figure supplement 3B*). These results suggest that WRN dependence of MSI-H cancer cell lines might not be attributable to an acute and hard-wired synthetic lethal interaction between WRN and MLH1/MSH3.

## CRISPR-Cas9-mediated inactivation of WRN confirms the selective dependency of MSI-H CRC models on WRN

We carried out CRISPR-Cas9 depletion assays in MSS and MSI-H CRC models to independently confirm the selective WRN dependencies observed in shRNA/siRNA studies. Cell lines were stably transduced with Cas9 followed by transduction of lentiviral particles co-expressing GFP and single guide RNAs (sgRNAs) targeting WRN or the essential replication factor RPA3 (*Figure 3A*). To investigate the relevance of the different protein domains in WRN by CRISPR scanning (*Shi et al., 2015*),

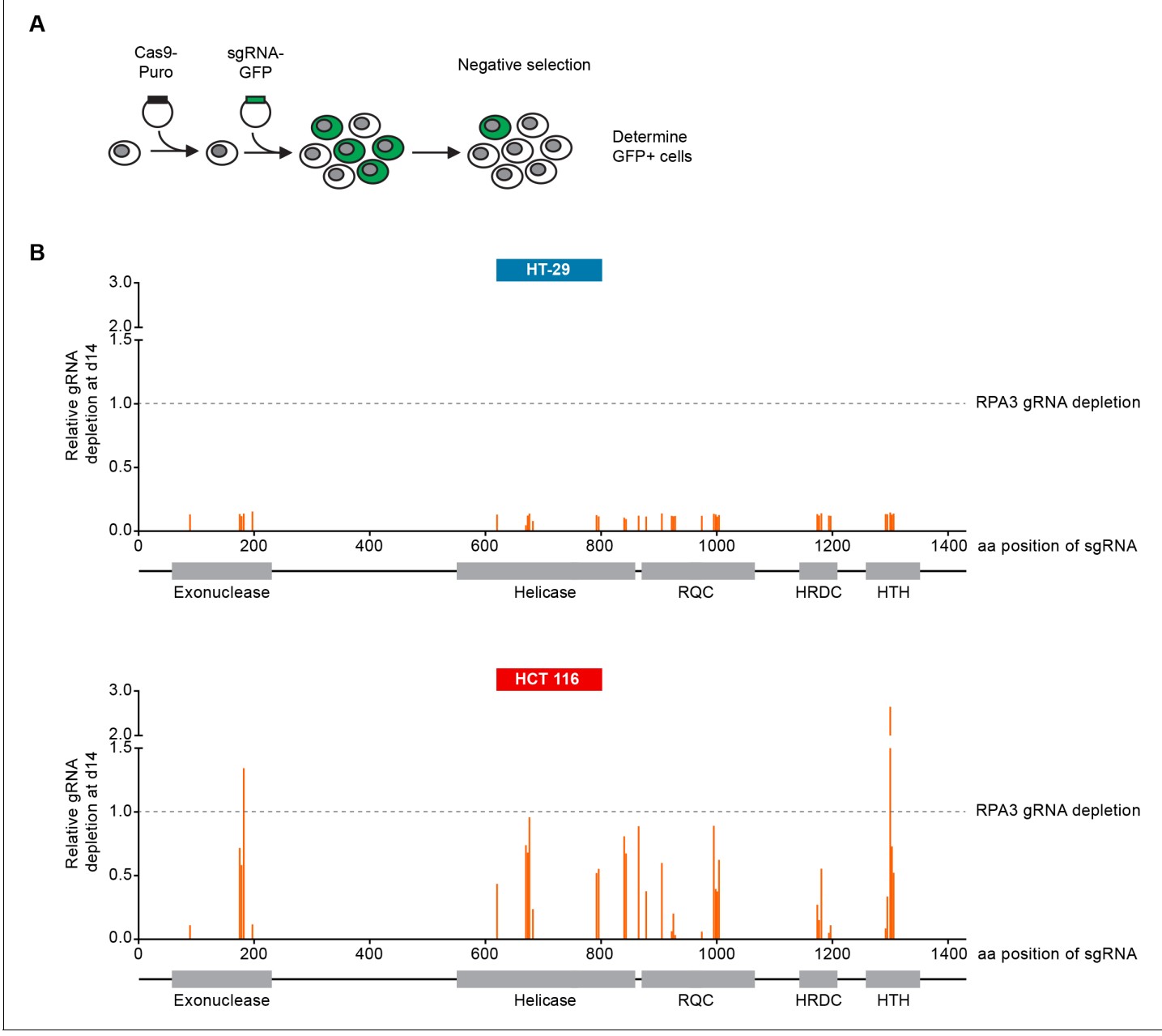

**Figure 3.** CRISPR-Cas9-mediated knock-out of WRN confirms the selective dependency of MSI-H CRC models on WRN. (**A**) Schematic representation of CRISPR-Cas9 depletion assays. Cas9 expressing cells were transduced with a lentivirus encoding GFP and sgRNAs. The percentage of GFP-positive cells was determined over time by flow cytometry. (**B**) Cas9 expressing MSS or MSI-H CRC cells were transduced with a lentivirus encoding GFP and sgRNAs targeting multiple domains in WRN as indicated. The percentage of GFP-positive cells was determined 14 days post-transduction and normalized to the fraction of GFP-positive cells at the first measurement. Depletion ratios are shown relative to the positive control RPA3 (n = 1 experimental replicate). Domains are annotated according to PFAM entry Q14191. RQC, RecQ helicase family DNA-binding domain; HRDC, Helicase and RNase D C-terminal, HTH, helix-turn-helix motif.

DOI: https://doi.org/10.7554/eLife.43333.008

The following figure supplement is available for figure 3:

**Figure supplement 1.** CRISPR-Cas9-mediated knock-out of WRN confirms the selective dependency of MSI-H CRC models on WRN.

DOI: https://doi.org/10.7554/eLife.43333.009

domain specific sgRNAs were used to target the exonuclease, helicase, RecQ helicase family DNA-binding (RQC) and Helicase and RNase D C-terminal (HRDC) domains and the C-terminal helix-turn-helix (HTH) motif. Negative selection of sgRNA expressing cells was monitored over 14 days via flow cytometry-based quantification of GFP expressing cells, and normalized to the effect of the RPA3 positive control sgRNA (*Figure 3B*). We did not observe depletion of cells harboring WRN targeting sgRNAs in the MSS CRC cell line HT-29. In contrast, MSI-H HCT 116 cells expressing WRN sgRNAs depleted to a similar level as observed for the RPA3 positive control. sgRNAs directed against the exonuclease, helicase and RQC domains were most effective, implying a functional or structural requirement of both these domains in the context of WRN dependency. Interestingly, strong depletion effects were also observed for sgRNAs targeting the C-terminal HTH motif (*Figure 3B*). A similar WRN sgRNA depletion pattern was observed in the MSI-H CRC cell line RKO, while we found far less pronounced depletion effects in the MSS CRC model SK-CO-1 (*Figure 3—figure supplement 1*). In agreement with the RNAi studies, our CRISPR-Cas9 experiments suggest that WRN provides an essential gene function in two MSI-H CRC cell lines but not in two MSS CRC lines.

## WRN dependency in MSI-H CRC is linked to its helicase function

To further dissect the relevance of WRN exonuclease and helicase function in WRN-dependent cell models, we generated FLAG-tagged, siRNA-resistant WRN (WRNr) expression constructs harboring loss-of-function mutations within the exonuclease- (E84A, Nuclease-dead) and helicase-domain (K577M, ATP-binding deficient), or both domains (E84A/K577M, Double-mutant) (*Gray et al., 1997*; *Huang et al., 1998*) (*Figure 4A*). Wild-type and mutant forms of WRNr were transduced in HCT 116 cells and monoclonal lines with matched stable WRNr expression were generated. Expression and nuclear accumulation of transgenic WRNr variants among the cell lines was confirmed using immunofluorescence analysis (*Figure 4B*). Immunoblotting revealed that transgenic WRNr wild-type as well as the mutant WRN proteins were expressed at levels higher than the endogenous WRN counterpart (*Figure 4B*). Two wild-type WRNr expressing clones were selected based on the respective high and low expression of the transgene in order to cover the range of mutant WRNr variant expression observed in the selected panel of clones. As expected, viability of empty vector control-transduced cells was strongly reduced upon depletion of WRN (*Figure 4C*). Importantly, both the high and low expression level of wild-type WRNr was sufficient to render HCT 116 cells inert to knock-down of endogenous WRN (*Figure 4C*). This demonstrates the on-target effect of the WRN siRNA duplex and indicates that the transgene-mediated rescue of WRN function is not protein level sensitive. Exogenous expression of the nuclease-dead form of WRNr almost completely rescued the effect of endogenous WRN depletion. In stark contrast, although expressed at similar or higher levels than WRNr wild-type, both the ATP-binding deficient and double-mutant form of WRNr were unable to restore viability following depletion of endogenous WRN (*Figure 4C*). In RKO cells expressing WRNr variants, we observed a slightly stronger dependency on WRN helicase ATP-binding function compared to exonuclease activity upon knock-down of endogenous WRN (*Figure 4—figure supplement 1A and B*), while WRNr-expressing monoclonal lines generated from the MSI endometrial carcinoma model HEC-265 showed an exclusive dependency on WRN helicase function (*Figure 4—figure supplement 1C*), similar to HCT 116 cells. In summary, these results indicate that the ATP-binding activity of WRN and possibly its helicase function are crucial for the survival of MSI-H CRC cells.

## Loss of WRN causes mitotic defects and nuclear abnormalities in MSI-H cells

In order to investigate the cellular basis for the viability reduction of MSI-H cancer cells upon WRN depletion, we monitored the consequences of WRN loss-of-function using immunofluorescence analysis of the nuclear membrane protein LAP2β and Hoechst DNA staining in MSS and MSI-H CRC cell lines. SW480 did not display any phenotypic differences upon transfection with NTC and WRN-targeting siRNAs (*Figure 5A*). Strikingly, in HCT 116 and RKO cells we observed formation of chromatin bridges and micronuclei upon WRN knock-down, both potential consequences of failed sister genome partitioning during mitosis (*Figure 5A*). Enlarged nuclei, indicative of failed mitosis, were additionally observed upon WRN knock-down in HCT 116 cells. Quantification of the frequency of chromatin bridges and micronuclei revealed that WRN depletion did not affect baseline levels of these aberrant nuclear morphologies in the MSS CRC cancer models SK-CO-1 and SW480, while the

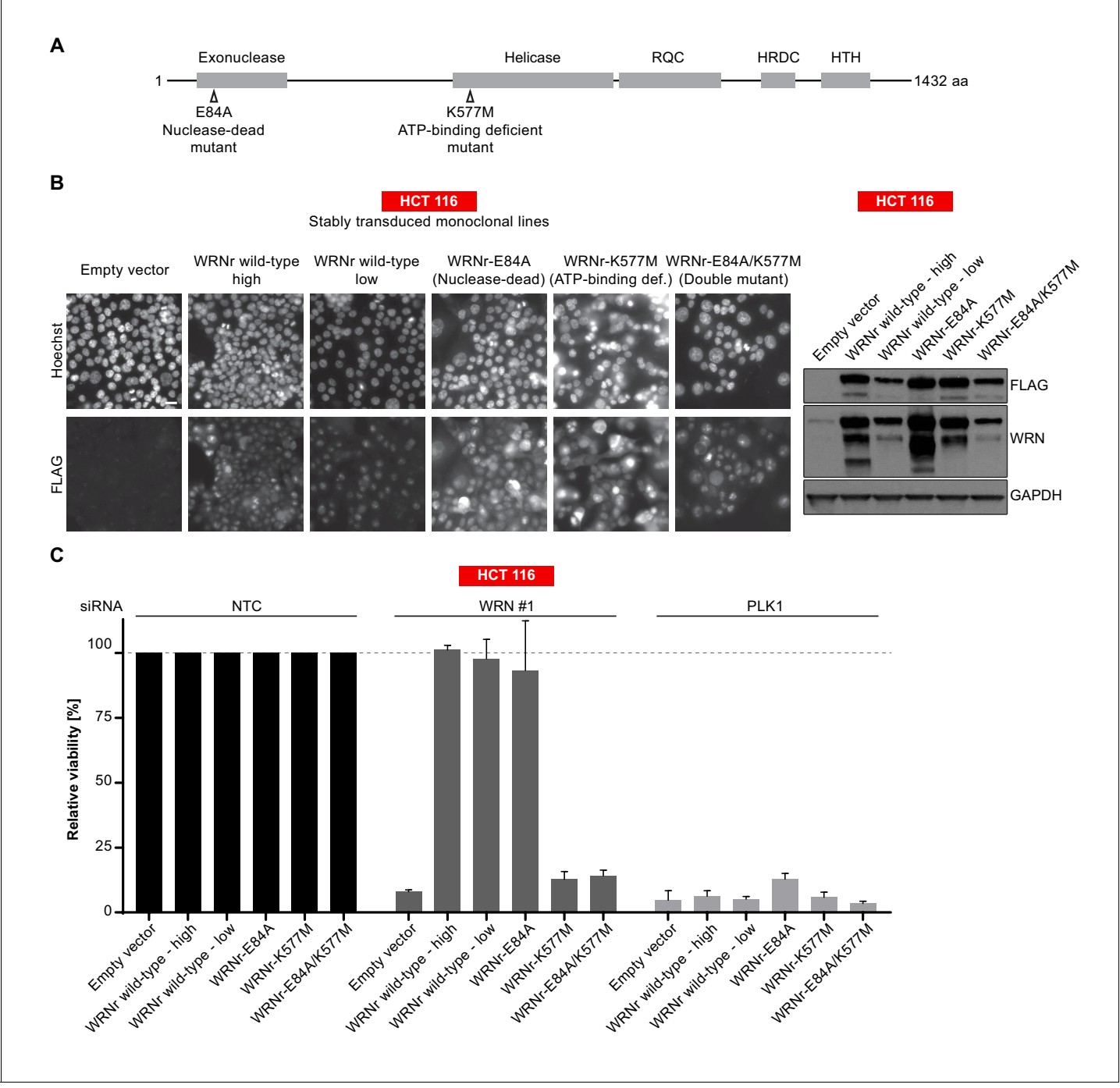

**Figure 4.** WRN dependency in MSI-H cancer cell lines is linked to its helicase function. (A) Schematic representation of WRN domain structure. Location of nuclease- and ATPase-inactivating mutations in siRNA-resistant WRN (WRNr) expression constructs is indicated. (B) MSI-H CRC HCT 116 cells were stably transduced with FLAG-tagged wild-type or mutant forms of WRNr and monoclonal lines with similar WRNr expression levels were isolated. For WRNr wild-type, two clones with high and low transgene expression were selected to cover the expression range of WRNr variants. Anti-FLAG immunofluorescence analysis was performed to monitor homogenous expression of WRNr. Expression of WRNr wild-type and mutant forms and endogenous protein levels were determined using immunoblotting with anti-FLAG and anti-WRN antibodies. GAPDH expression was used to monitor equal loading. Scale bar, 20 μM. (C) WRNr-expressing HCT 116 cells were transfected with the indicated siRNAs. Cell viability was determined 7 days after transfection and is shown relative to NTC siRNA. Data information: In (C), data are presented as mean ± SD of three independent experiments.
DOI: https://doi.org/10.7554/eLife.43333.010

The following figure supplement is available for figure 4:

**Figure supplement 1.** WRN dependency in MSI-H cancer cell lines is linked to its helicase function.

*Figure 4 continued on next page*

*Figure 4 continued*

DOI: https://doi.org/10.7554/eLife.43333.011

frequency of both chromatin bridges and micronuclei was strongly increased upon WRN knock-down in the MSI-H CRC cell lines HCT 116 and RKO (*Figure 5B and C*). In non-transformed hTERT RPE-1 cells, WRN depletion led to a slight increase of chromatin bridge and micronucleus formation, although far less pronounced compared to the MSI-H CRC models (*Figure 5B and C*). Supportive of the immunofluorescence studies, live cell imaging revealed a strong increase of the incidence of lagging chromosomes and chromosome bridges during mitosis in WRN-depleted MSI-H CRC cell lines HCT 116 and RKO, but not in MSS SW480 cells (*Figure 6*). We conclude that WRN depletion in MSI-H cells results in nuclear morphology and integrity aberrations that are manifested during cell division. The correlation of these defects with the observed cell viability reduction in MSI-H models suggests that nuclear abnormalities are causally linked to the anti-proliferative effect of WRN loss.

## Structural chromosome aberrations in MSI-H cells after WRN loss-of-function

The observed nuclear integrity and mitotic defects caused by WRN depletion in MSI-H cells could be the consequence of preceding genome maintenance aberrations. This hypothesis is reinforced by the important role of RECQ family helicases, including WRN, in genome integrity (*Chu and Hickson, 2009*). To interrogate genome integrity, we performed mitotic chromosome spread analysis in MSI-H, MSS and non-transformed cells after depletion of WRN. To overcome the low abundance of mitotic cells in WRN-depleted MSI-H cell lines, caffeine was added to cultured cells to bypass the G2/M checkpoint. Strikingly, WRN loss elicited structural chromosome aberrations, such as chromosome breaks and non-homologous radial formations, in the MSI-H CRC cell lines HCT 116 and RKO (*Figure 7A*). Quantification of the number of chromatin breaks per karyotype demonstrated a strong increase in the fraction of cells harboring 1–5 or >5 chromosome breaks in the two MSI-H CRC models upon WRN depletion (*Figure 7B*). In the MSS CRC model SW480 and hTERT RPE-1 non-transformed cells only a minor increase of karyotypes with chromosome breaks was detected (*Figure 7B*).

To test whether signs of DNA damage can already be detected in WRN-depleted interphase cells, we analyzed the intensity of nuclear γ-H2AX signals. We observed a strong induction of γ-H2AX upon siRNA-mediated depletion of WRN in MSI-H, but not MSS, CRC cancer cell lines (*Figure 7—figure supplement 1*). Strikingly, in both HCT 116 and RKO MSI-H CRC lines, WRN depletion led to an increase in nuclear γ-H2AX signal similar to or higher than upon treatment of cells with the DNA double-strand break-inducing agent etoposide (*Figure 7—figure supplement 1A and B*). Importantly, nuclear γ-H2AX levels in non-transformed and MSS hTERT RPE-1 cells were strongly increased by treatment with etoposide, while loss of WRN had no effect in this model (*Figure 7—figure supplement 1B*).

Our analyses suggest that WRN helicase is essential for maintaining genome integrity in MSI-H cells by preventing chromosome breaks and erroneous chromosome fusions. The observed mitotic chromosome aberrations in WRN-depleted MSI-H cells can also explain the aforementioned nuclear morphology and mitotic defects, including micronuclei, lagging chromosomes and chromatin bridges. The detection of DNA damage markers in interphase nuclei of WRN-depleted MSI-H cells suggests that the aforementioned mitotic and post-mitotic aberrations originate from genome integrity defects in interphase. The correlation of chromosome aberrations and nuclear abnormalities with MSI-H status following loss of WRN function indicate that genome integrity defects are responsible for the profound reduction in viability of MSI-H cancer cells.

## Discussion

Treatment paradigms for MSI-H tumors have recently shifted with the approval of the immune checkpoint agents pembrolizumab, nivolumab and ipilimumab, targeting programmed cell death 1 (PD-1) and cytotoxic T-lymphocyte-associated Protein 4 (CTLA-4) in this patient segment (*Le et al., 2017*; *Le et al., 2015*; *Overman et al., 2018*). Pembrolizumab constitutes the first cancer therapy approval based on a patient selection biomarker irrespective of the tumor type, highlighting MSI-H status as a therapeutically trackable and clinically implemented feature of tumor cells (*Goswami and*

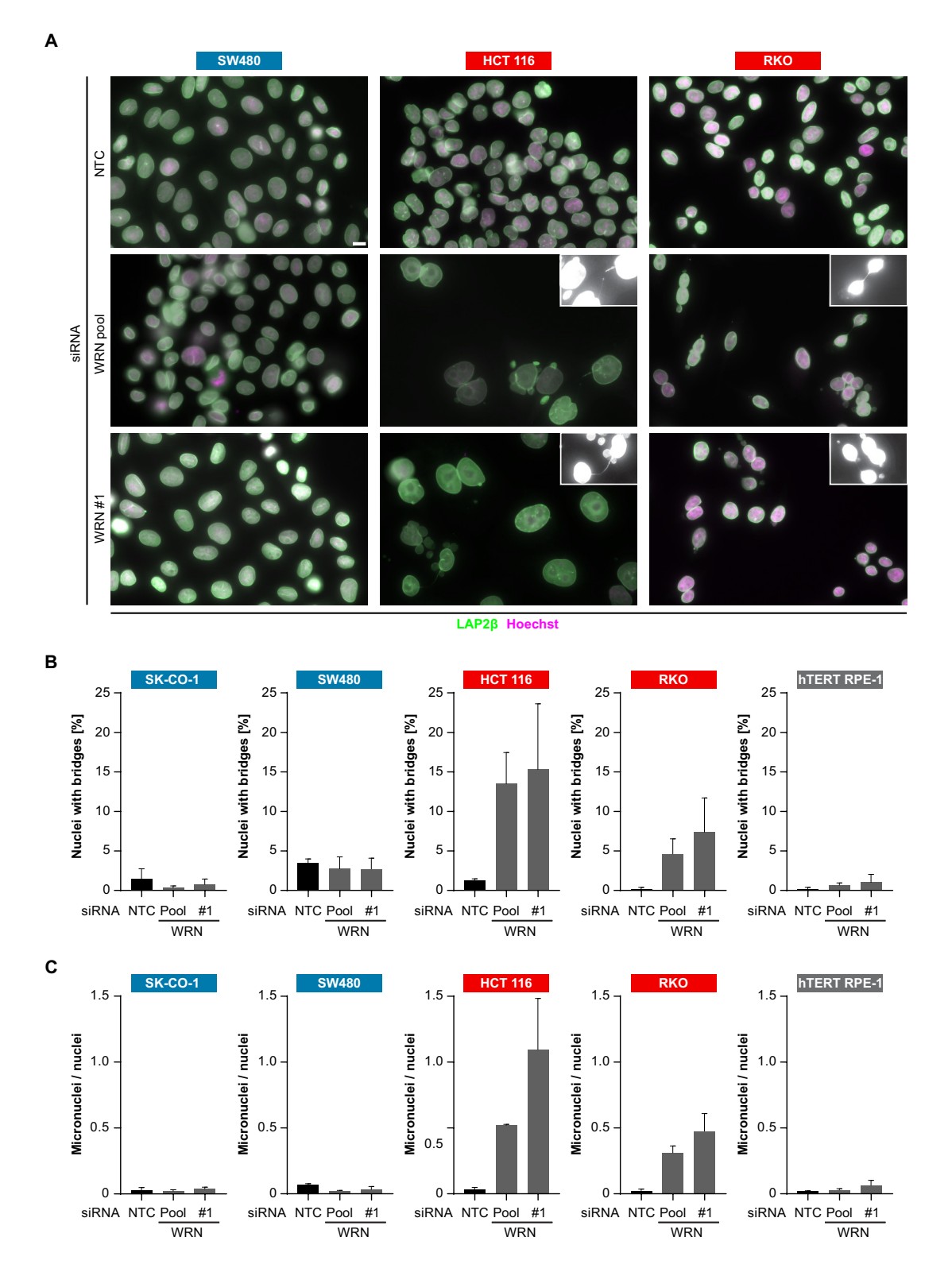

**Figure 5.** WRN loss-of-function in MSI-H CRC results in nuclear morphology and integrity defects. (**A**) MSS and MSI-H CRC cell lines were transfected with the indicated siRNAs. Immunofluorescence analysis was performed 96 hr after transfection to determine the fraction of cells with chromosomal bridges and micronuclei. Examples with enhanced brightness are shown as insets. LAP2β signal intensity was adjusted in a subset of samples for uniform representation. Scale bar, 10 μm. (**B**) Statistical analysis of chromosomal bridge phenotypes observed in siRNA knock-down studies in MSS and
*Figure 5 continued on next page*

*Figure 5 continued*

MSI-H CRC lines and hTERT RPE-1 non-transformed cells. (**C**) Statistical analysis of micronuclei phenotypes observed in siRNA knock-down studies in MSS and MSI-H CRC lines and hTERT RPE-1 non-transformed cells. Data information: In (**B** and **C**), data are presented as mean ± SD of two or three independent experiments (n ≥ 410 cells).

DOI: https://doi.org/10.7554/eLife.43333.012

*Sharma, 2017*). While responses to immune checkpoint blockade in MSI-H cancer are often durable, intrinsic and acquired resistance to immunotherapy represents a continuous medical need in MSI-H cancer.

The highest prevalence of MMR-deficiency/MSI-H status is observed among CRC, gastric and endometrial cancers (*Cortes-Ciriano et al., 2017*). Recent genomic analyses have outlined a detailed landscape of genomic alterations in MSI-H tumors (*Cancer Genome Atlas Network, 2012*; *Knijnenburg et al., 2018*). However, while MSI-H status is linked to a CpG island methylator (CIMP) phenotype and an increased mutational burden, a functional dependency enabling selective targeting of MSI-H cancer cells remains elusive. The results of this study uncover a novel vulnerability of MSI-H tumor cell models and indicate that pharmacological inhibition of WRN ATPase/helicase function might serve as an attractive targeted therapeutic strategy in MSI-H cancer.

Our data suggest that similar to the tumor agnostic activity of checkpoint blockade, MMR deficiency and MSI-H status represent a genetic determinant for WRN dependency regardless of tumor type. As indicated by the MLH1/MSH3 reconstitution and depletion studies, the genetic interaction of WRN and MMR genes cannot readily be recapitulated using isogenic cell models and thus seems to be distinct from acute and hard-wired synthetic lethal interactions, such as described in cancer cells for the alternate BAF complex ATPases SMARCA2/SMARCA4 or the cohesin subunits STAG1/STAG2 (*Oike et al., 2013*; *van der Lelij et al., 2017*; *Benedetti et al., 2017*). In the Random Forest model (see *Figure 1—figure supplement 1A*), loss of MLH1 expression was the feature most strongly associated with WRN dependency. However, we cannot rule out that alterations frequently co-occurring with MMR-deficiency or MSI-H status (*Boland and Goel, 2010*; *Knijnenburg et al.,*

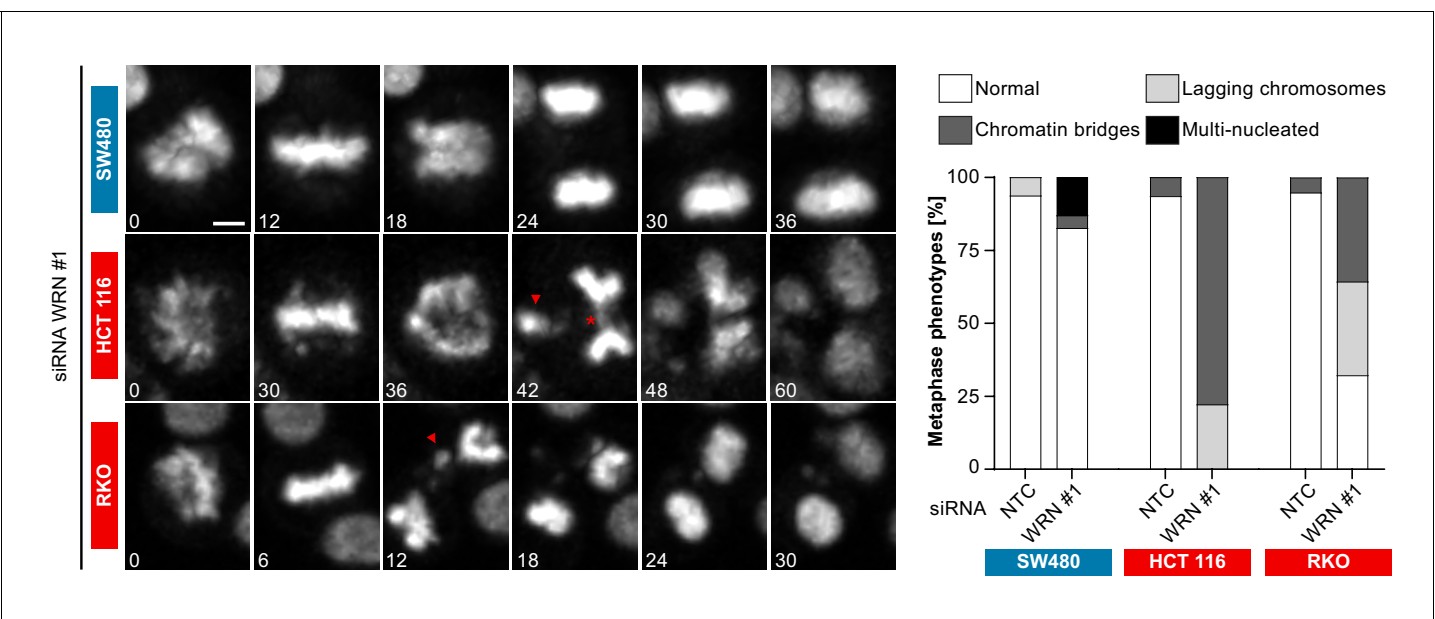

**Figure 6.** Time-lapse analysis of mitosis in WRN-depleted MSS and MSI-H CRC models. Mitotic live cell imaging in WRN-depleted MSS and MSI-H CRC cell lines. Cells were transfected with WRN siRNA #1. Cells were stained with SiR-Hoechst dye and were analyzed 24 hr post siRNA transfection. Exemplary lagging chromosomes (arrowhead) and a chromatin bridge (asterisk) are designated. Duration of time-lapse is indicated in minutes. Scale bar, 5 μM.

DOI: https://doi.org/10.7554/eLife.43333.013

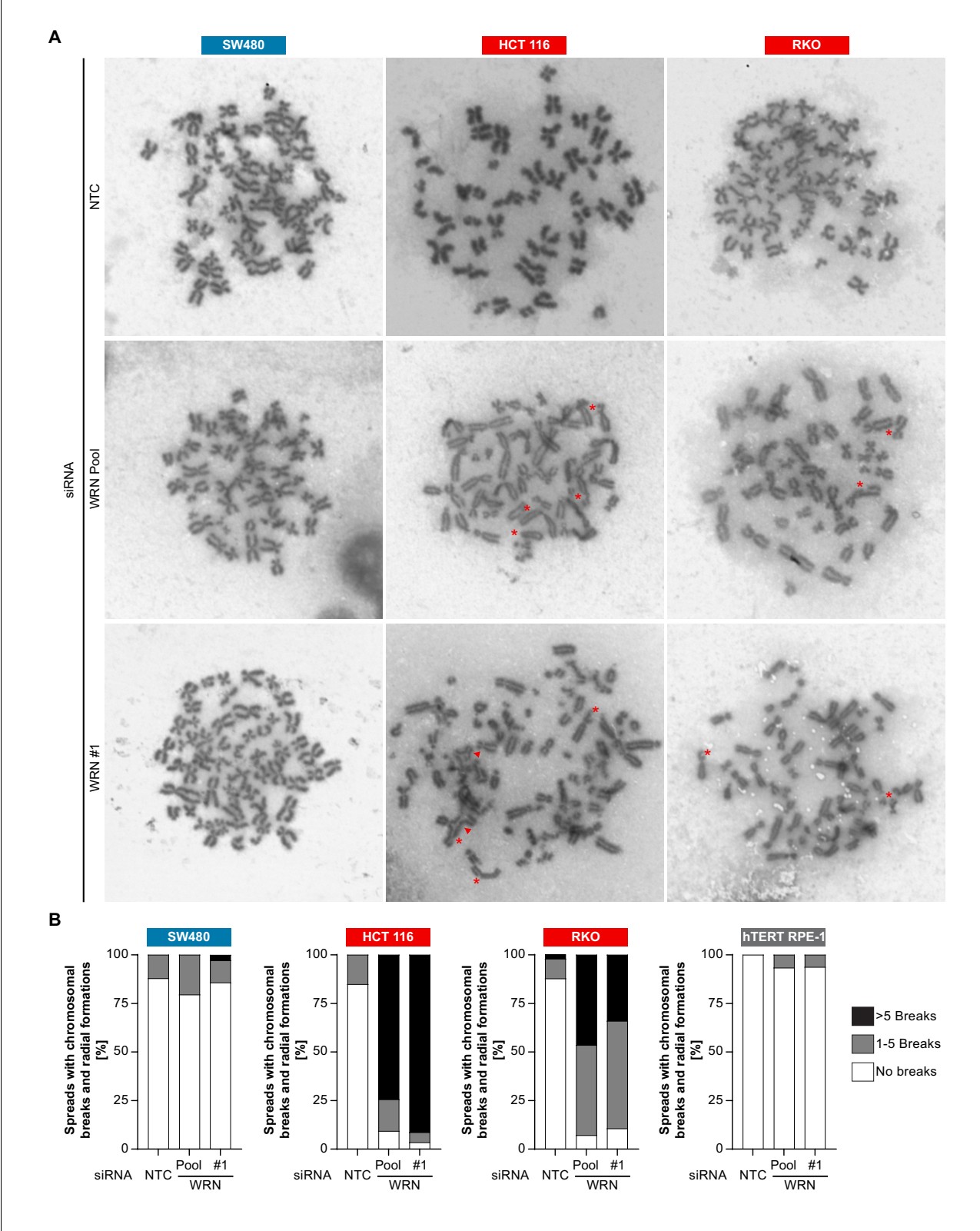

**Figure 7.** Loss of WRN function in MSI-H CRC causes severe chromosomal defects. (**A**) MSS and MSI-H CRC cell lines were transfected with the indicated siRNAs. Mitotic chromosome spreads were prepared 72 hr after transfection and visualized by microscopy. Non-homologous radial formations are designated by arrows, breaks are labeled with asterisks. (**B**) Quantification of chromosomal defects observed in siRNA knock-down studies in MSS and MSI-H CRC lines and hTERT RPE-1 non-transformed cells. The status of chromosomal breaks of individual metaphase spreads was

*Figure 7 continued on next page*

*Figure 7 continued*
categorized into normal, 1–5 breaks or more than five breaks (n ≥ 28 spreads of two independently analyzed slides). Non-homologous radial formation was counted as two breaks.
DOI: https://doi.org/10.7554/eLife.43333.014
The following figure supplement is available for figure 7:
**Figure supplement 1.** Elevated levels of the DNA damage marker γ-H2AX upon loss of WRN function in MSI-H CRC cells.
DOI: https://doi.org/10.7554/eLife.43333.015

*2018*) alone or in combination contribute to the context-dependent requirement on WRN function in MSI-H cancer cell lines. Further work is required to decipher which specific or cumulative genetic alterations elicit WRN dependency in MMR-deficient and MSI-H cells.

Upon WRN loss-of-function we observe a strong and rapid decrease in the viability of MSI-H cell models that is accompanied by nuclear abnormalities and cell division defects. In particular, we find that WRN-depleted MSI-H cancer cells display chromosome breaks, chromatin bridges and micronuclei indicative of genome instability that is highlighted during cell division. The strong induction of γ-H2AX in WRN-depleted MSI-H interphase cells suggest that pervasive DNA damage, e. g. DNA double strand breaks, accumulate already in interphase prior to entry into mitosis. The occurrence of these defects and phenomena in MSI-H but not MSS cells upon WRN inactivation suggests that these aberrations might be causally linked to the selective reduction in viability in MSI-H cells. While rescue studies using WRN variants clearly indicate WRN helicase function as the critical enzymatic activity in MSI-H cell models, CRISPR domain scanning indicates a potential structural requirement of the exonuclease domain and the HTH loop of WRN.

WRN is a member of the RecQ helicase family which fulfils pleiotropic functions in DNA repair (*Chu and Hickson, 2009*). MMR activity is required for activation of the G2/M checkpoint in response to DNA damage prior to entry into mitosis (*O'Brien and Brown, 2006*). WRN function in MSI-H cells might therefore be critical for the resolution of DNA damage events and to prevent premature entry into mitosis. Of note, cell lines derived from Werner syndrome patients display defective mitotic recombination and are susceptible to genome instability (*Prince et al., 2001*). However, in MSS cancer and non-transformed cells WRN depletion had no or very mild effects on viability, suggesting that pharmacological inhibition of WRN might allow for an MSI-H cancer-directed therapy that spares normal cells and tissues.

The chromosome breaks and radial chromosomes observed in MSI-H cells upon WRN depletion indicate the generation and/or persistence of DNA double strand breaks. Future research is required to dissect the molecular basis for this effect. It is conceivable that WRN is required to process and resolve DNA repair or replication intermediates that arise in MMR-deficient cells or that MMR is required to cope with intermediates emerging upon compromised WRN function. The identification of the molecular basis of the WRN-MSI-H relationship will also help to understand why some rare outlier MSI-H cell lines (see *Figure 1B*) do not respond to WRN inactivation.

Werner syndrome patients show an increased lifetime risk to develop tumors, pointing to a tumor-suppressive function of WRN (*Goto et al., 2013*). Interestingly, homozygous *Wrn*-null mice display no overt phenotype and do not recapitulate the premature ageing or cancer predisposition conditions of Werner syndrome, unless crossed into a *Terc*-null background (*Chang et al., 2004*; *Lebel and Leder, 1998*; *Lombard et al., 2000*). Mutations in Werner syndrome are almost exclusively truncating nonsense, splicing or frameshift mutations affecting WRN nuclear localization, suggesting that concomitant loss of WRN helicase and exonuclease function might be required for the onset of Werner syndrome (*Fu et al., 2017*; *Huang et al., 2006*; *Matsumoto et al., 1997*; *Yokote et al., 2017*). This indicates that inhibition of WRN helicase function might have a therapeutic index for the treatment of MSI-H cancer without inducing Werner syndrome related phenotypes.

Our study highlights the power of combining deep functional genomic screening data with tumor cell line profiling to identify new targets with an associated predictive biomarker in oncology. Given the possibility to develop potent and selective small molecule inhibitors of WRN helicase (*Aggarwal et al., 2013*; *Aggarwal et al., 2011*; *Rosenthal et al., 2010*), our findings outline a novel strategy for the treatment of a clinically defined subset of patients harboring MSI-H/MMR-deficient tumors. Since genome instability can elicit cytoplasmic nucleic acid sensor pathways and innate

immune responses (*Mackenzie et al., 2017*), the induction of cancer cell selective nuclear aberrations by WRN inactivation could provide a synergistic combination option with the approved immunotherapy agents for the benefit of MSI-H cancer patients.

# Materials and methods

**Key resources table**

| Reagent type (species) or resource | Designation | Source or reference | Identifiers | Additional information |
|---|---|---|---|---|
| Gene (*Homo sapiens*) | Werner Syndrome RecQ Like Helicase (WRN) | N/A | Entrez Gene: 7486 | |
| Genetic reagent (*Homo sapiens*) | NTC siRNA | Dharmacon | D-001810–10 | non-targeting siRNA pool |
| Genetic reagent (*Homo sapiens*) | WRN siRNA pool | Dharmacon | L-010378–00 | WRN-targeting siRNA pool |
| Genetic reagent (*Homo sapiens*) | WRN siRNA | Dharmacon | J-010378–05 | WRN-targeting siRNA |
| Cell line (*Homo sapiens*) | HCT 116 | ATCC | RRID:CVCL_0291 | MSI-H CRC cell line |
| Cell line (*Homo sapiens*) | RKO | ATCC | RRID:CVCL_0504 | MSI-H CRC cell line |
| Cell line (*Homo sapiens*) | SW480 | ATCC | RRID:CVCL_0546 | MSS CRC cell line |
| Cell line (*Homo sapiens*) | SK-CO-1 | ATCC | RRID:CVCL_0626 | MSS CRC cell line |
| Cell line (*Homo sapiens*) | hTERT RPE-1 | ATCC | RRID:CVCL_4388 | Non-transformed telomerase immortalized cell line |
| Antibody | mouse anti-WRN | Cell Signaling | RRID:AB_10692114 | 1/1000 dilution for immunoblot |
| Antibody | mouse anti-GAPDH | Abcam | RRID:AB_2107448 | 1/30000 dilution for immunoblot |
| Antibody | mouse anti-FLAG | SIGMA | RRID:AB_262044 | 1/1000 dilution for immunoblot |
| Antibody | mouse anti-LAP2ß | BD Transduction Laboratories | RRID:AB_398313 | 1/100 for immunofluorescence analysis |
| Recombinant DNA reagent | pLVX-WRN-3x FLAG-IRES-Puro | This study | | Lentiviral vector for stable expression of WRN wild-type |
| Recombinant DNA reagent | pLVX-WRN-3x FLAG-IRES-Puro E84A | This study | | Lentiviral vector for stable expression of WRN E84A mutant |
| Recombinant DNA reagent | pLVX-WRN-3x FLAG-IRES-Puro K577M | This study | | Lentiviral vector for stable expression of WRN K577M mutant |
| Recombinant DNA reagent | pLVX-WRN-3x FLAG-IRES-Puro E84A _K577M | This study | | Lentiviral vector for stable expression of WRN E84A/K577M double mutant |
| Other | Drive database | PMID 28753431 | | Functional genomics database on cancer cell line dependencies |

## Random Forest model

To explore the hypothesis that WRN sensitivity is associated with MSI, we employed an exploratory machine learning approach (*Qi, 2012*). We first divided the DRIVE WRN cell line sensitivity data (*McDonald et al., 2017*) into four distinct groups, using a k-means clustering algorithm. We chose

four clusters to model insensitive (cluster 1), moderate insensitive (cluster 2), moderate sensitive (cluster 3) and sensitive (cluster 4) cell lines. We subsequently denoted clusters 1 and 2 as insensitive and clusters 3 and 4 as sensitive and chose an RSA score $<-1.37$ (maximum value of cluster 3) as a cutoff between sensitive and insensitive lines. Only a small fraction of cell lines, 30 out of 371 cell lines (8%), is sensitive in the entire data set, suggesting a pronounced class imbalance between sensitive and insensitive cell lines. To address this class imbalance, we focused our subsequent analysis on cell lines originating from CRC, as i) most sensitive cell lines are from this indication and ii) MSI has been extensively characterized in this indication (*Boland and Goel, 2010*; *Medico et al., 2015*). 36% (13 out of 36) of colon cancer cell lines are sensitive to WRN loss of function according to our k-means clustering based approach.

We next assembled a MSI feature list. We used cell line gene expression and mutation data for the cell lines from a set of genes, i) involved in MMR (EXO1, MLH1, MLH3, MSH2, MSH3, MSH6) and ii) genetic target genes of MSI in CRC (*Boland and Goel, 2010*). We next trained a Random Forest model based on 50% of the data. On the full dataset, the model classifies WRN sensitive and insensitive cell lines with an accuracy of 0.89 and a recall rate for sensitive lines of 0.69.

## MSI analysis

Genomic DNA was isolated using QIAampDNA mini kit (Qiagen, Hilden, Germany). Per reaction 2 ng of genomic DNA was used for fluorescent PCR-based analysis of the mononucleotide microsatellite marker length using the Promega MSI Analysis System, Version 1.2 kit. Microsatellite fragment length of a standard panel of markers (NR-21, BAT-26, BAT-25, NR-24 and MONO-27) according to the 'Revised Bethesda Guidelines for Hereditary Nonpolyposis Colorectal Cancer (Lynch Syndrome) and Microsatellite Instability' (*Umar et al., 2004*) was analyzed using capillary electrophoreses (Applied Biosystems 3130xl Genetic Analyzer, 16-capillary electrophoresis instrument) and evaluated with GeneMapper Software 5 (Applied Biosystems).

## Cell culture and lentiviral transduction

HCT 116 cells were cultured in McCoy's 5A medium (GIBCO, 36600–021) with glutamax supplemented with 10% fetal calf serum (FCS), hTERT RPE-1 cells were cultured in DMEM:F12 (ATCC, 30–2006) supplemented with 10% FCS and 0.01 mg/ml Hygromycin B. RKO, SW480, CaCo-2 and SK-CO-1 cells were cultured in EMEM (SIGMA, M5650) with glutamax supplemented with 10% FCS and Na-Pyruvate. SNU-C4 cells were cultured in RPMI1640 medium (ATCC, 30–2001) with glutamax, supplemented with 10% FCS, 25 mM HEPES and 25 mM NaHCO3. LS1034 cells were cultured in RPMI-1640 (ATCC, 30–2001) supplemented with 10% FCS. MFE-280 cells were cultured in 40% RPMI 1640 (GIBCO, 61870-010), 40% DMEM (SIGMA, D6429) supplemented with 20% FCS and 1X insulin-transferrin-sodium selenite (GIBCO, 41400–045). HEC-265 cells were cultured in EMEM (SIGMA, M5650) with glutamax supplemented with 15% FCS. ISHIKAWA cells were cultured in EMEM (SIGMA, M5650) with glutamax medium supplemented with 5% FCS and NEAA. HEC-6 cells were cultured in EMEM (SIGMA, M5650) with glutamax supplemented with 15% FCS, NEAA and Na-Pyruvate. HT-29_CRISPR-Cas9 cells were cultured in McCoy's 5A (GIBCO, 36600–021) with glutamax supplemented with 10% FCS and 10 µg/ml Blasticidin (Invitrogen, R210-01). HCT 116_CRISPR-Cas9 cells were cultured like the parental cell line but supplemented with 2 µg/ml Puromycin. All supplements were obtained from GIBCO, FCS (SH30071.03) from GE Healthcare Life Sciences and Puromycin from SIGMA (P9620). Lentiviral particles were produced using the Lenti-X Single Shot system (Clontech, Mountain View, CA, US). Following lentiviral infection, stably transduced pools were generated using Puromycin selection (HCT 116: 2 µg/ml, RKO: 0.5 µg/ml, SK-CO-1: 1 µg/ml) or Blasticidin (HT-29: 10 µg/ml). Sources, MSI status and authentication information (STR fingerprinting at Eurofins Genomics, Germany) of cell lines used in this study are provided in *Supplementary file 2*. All cell lines were tested negatively for mycoplasma contamination and have been authenticated by STR fingerprinting (HCT 116 + ch2, HCT 116 + ch3, HCT 116 + ch3+ch5 and subclones generated from SW480 were not analyzed by STR fingerprinting).

## siRNA transfection and cell viability

For knock-down experiments, cells were transfected with ON-TARGETplus SMARTpool siRNA duplexes (Dharmacon, Lafayette, CO, US) targeting WRN (L-010378–00), MLH1 (L-003906–00) or

MSH3 (L-019665–00) using Lipofectamine RNAiMAX reagent according to the manufacturer's instructions (Invitrogen, Waltham, MA, US). For WRN knock-down, additionally an individual siRNA was used (J-010378–05). Chromosome spreads, immunoblotting, immunofluorescence and live cell imaging experiments were performed using a final siRNA concentration of 20 nM. Cell viability assays were performed using 10 nM siRNA in 96-well plates in a total volume of 100 µl per well. Viability was determined using CellTiter-Glo (Promega, Madison, WI, US). Seven days post-transfection, 100 µl CellTiter-Glo solution was added directly to the cell medium, mixed and incubated for 10 min prior to determination of the luminescence signal. Crystal violet staining was performed in 24-well format using a total volume of 1000 µl per well. Seven days post-transfection, cells were fixed with 4% formaldehyde in PBS and stained twice for 30 min with crystal violet (SIGMA, HT901). For co-depletion of p53 and WRN 10 nM of the respective siRNA duplexes each were used for immunoblot and viability assay.

## Cell extracts for immunoblotting
Cell pellets were resuspended in extraction buffer (50 mM Tris-HCl pH 8.0, 150 mM NaCl, 1% Nonidet P-40 supplemented with Complete protease inhibitor mix [Roche, Switzerland] and Phosphatase inhibitor cocktails [SIGMA, P5726 and P0044]).

## Antibodies
The following antibodies were used: WRN (8H3) mouse mAb (Cell Signaling, 4666, 1/1000 dilution), mouse anti-GAPDH (Abcam, ab8245, 1/30000 dilution), mouse anti-FLAG (SIGMA, F1804, 1/1000 [immunoblot] or 1/500 [immunofluorescence] dilution), mouse anti-LAP2ß (BD Transduction Laboratories #611000, 1/100 dilution), mouse anti-p53 (Calbiochem, OP43, 1/1000 dilution), rabbit anti-Actin (SIGMA, A2066, 1/4000) dilution), mouse anti-MLH1 (Cell Signaling, 3515, 1/1000 dilution), rabbit anti phospho-Histone H2A.X (Ser139) (Cell Signaling, 2577, 1/800 dilution) a rabbit anti-MSH3 (Santa Cruz, sc-11441, 1/500 dilution) and secondary rabbit (Dako, P0448, 1/1000 dilution), mouse anti-IgG-HRP (Dako, P0161, 1/1000 dilution) and mouse Alexa Fluor 488 (Molecular Probes, Eugene, OR, US, 1/1000 dilution).

## Quantitative reverse transcription PCR (qRT-PCR)
RNA was isolated 472 hr post-transfection and reversely transcribed using SuperScript VILO kit (Thermo Scientific). All qPCR analyses were performed with the QuantiTect Multiplex PCR kit (Qiagen, Hilden, Germany) on a StepOne Real-Time PCR Sytem (Applied Biosytems) with a total of 45 cycles. Constitutive maintenance gene 18S rRNA (Applied Biosystems, Quencher VIC/MGB, 4319413E) and human WRN (Applied Biosystems, Quencher FAM/MGB-NFQ, 4331182), human MLH1 (Applied Biosystems, Quencher FAM/MGB-NFQ, 4453320) and human MSH3 (Applied Biosystems, Quencher FAM/MGB-NFQ, 4448892) TaqMan probes were used. WRN expression was normalized to 18S rRNA expression levels and is indicated relative to the NTC control.

## CRISPR-Cas9-mediated gene knock-out
To introduce mutations in WRN in SW480, the following sgRNA sequences were cloned into pSpCas9 BB-2A-GFP pX458 (GenScript, China): GGCCACCATTATACAATAGA (EN-domain_01), GCAGTTAAAAAGGCAGGTGT (EN-domain_02) and GTCTTGCCGATCAATATCGC (RQ-domain_01). Cells were transfected with X-tremeGENE 9 DNA Transfection Reagent (SIGMA, 6365779001), sorted after 48 hr for GFP positive cells and diluted to obtain single cell clones. WRN knock-out was monitored by immunoblotting.

## CRISPR depletion assays
Stable Cas9 expressing cell lines, using either Blasticidin or Puromycin as selection markers, were transduced with vectors encoding GFP and sgRNAs targeting different domains of WRN (*Supplementary file 3*). On day three post-transfection the fraction of GFP positive cells, measured via flow cytometry analysis, was set to 100%. All values were normalized to the control RPA3_1.3 for relative depletion ratio.

## cDNA transgene vectors

pLVX-WRN-3xFLAG-IRES-Puro, pLVX-WRN-3xFLAG-IRES-Puro K577M, pLVX-WRN-3xFLAG-IRES-Puro E84A and pLVX-WRN-3xFLAG-IRES-Puro E84A _K577M for siRNA-resistant transgene expression were generated by gene synthesis (GenScript, China) based on the WRN cDNA sequence NCBI NM_000553.5 followed by cloning into the parental pLVX vector (Clontech, Mountain View, CA, US). Codon optimization changes were introduced into WRN coding sequences to render the transgenes siRNA-resistant.

## Immunofluorescence

For immunofluorescence, 72 hr post siRNA transfection cells were fixed with 4% paraformaldehyde for 15 min, permeabilized with 0.2% Triton X-100 in PBS for 10 min and blocked with 3% BSA in PBS containing 0.01% Triton X-100. Cells were incubated with primary LAP2ß or phospho-Histone H2A.X (Ser139) and secondary antibody (Alexa 488, Molecular Probes, Eugene, OR, US), DNA was counterstained with Hoechst 33342 (Molecular Probes, Eugene, OR, US; H3570). Coverslips and chambers were mounted with ProLong Gold (Molecular Probes, Eugene, OR, US). Images were taken with an Axio Plan2/AxioCam microscope and processed with MrC5/Axiovision software (Zeiss, Germany). For quantification of γ-H2AX immunofluorescence, nuclei were identified based on Hoechst staining using segmentation in ImageJ 1.52a and corresponding γ-H2AX mean intensities of nuclei were determined.

## Chromosome spreads and live-cell imaging

For chromosome spread analysis, Nocodazole (1.5 µM final concentration) and 2 mM caffeine was added to the medium for 6 hr. Cells were harvested and hypotonically swollen in 40% medium/60% tap water for 5 min at room temperature. Cells were fixed with freshly made Carnoy's solution (75% methanol, 25% acetic acid), and the fixative was changed three times. For spreading, cells in Carnoy's solution were dropped onto glass slides and dried. Slides were stained with 5% Giemsa (Merck) for 4 min, washed briefly in tap water and air dried. For chromosome spread analysis two independent slides were scored blindly for each condition. Live-cell imaging was performed using Spinning Disk Confocal UltraView Vox Axio Observer equipped with Plan apochromat 20x/0.8 objective (Zeiss) and an electron-multiplying charge-coupled device 9100–13 camera (Hamamatsu Photonics). The microscope was controlled using Volocity software (Perkin-Elmer). DNA was counterstained with 100 nM SiR-Hoechst 3 hr before the start of imaging. At 24 hr post siRNA transfection cell nuclei were imaged in 2 z slice sections spaced 6 µm every 6 min for 48 hr. For the imaging, cells were seeded into glass bottom 24 well SensoPlate (Greiner) with imaging medium (phenol red free DMEM supplemented with 10% [vol/vol] FCS, L-glutamine 2 mM and 1% [vol/vol] penicillin-streptomycin). During live cell imaging, cells were maintained at 37°C in a humidified 5% $CO_2$ atmosphere.

## Acknowledgements

We would like to thank Susanne Stockinger, Vanessa Rössler, Jodie Grant and Christoph Reiser (all Boehringer Ingelheim RCV GmbH) for generation of Cas9-expressing cell lines and Thomas Lendl (Institute of Molecular Pathology, Austria) for the support of immunofluorescence quantification analysis. We are grateful to Christoph Gasche (Medical University of Vienna) for providing the MSH3 antibody and Stephen West (The Francis Crick Institute, UK) for helpful suggestions and advice. The Research Institute of Molecular Pathology (IMP) is supported by Boehringer Ingelheim. Research in the laboratory of JMP is funded by Boehringer Ingelheim and grants by the Austrian Research Promotion Agency (FFG 834223, FFG 852936) and the European Research Council H2020 (693949). KN is supported by an EMBO Long Term Fellowship (ALTF 1335–2016) and a HFSP fellowship (LT001527/2017). PVDL is a member of the Boehringer Ingelheim Discovery Research global postdoc program.

# Additional information

## Competing interests

Simone Lieb, Silvia Blaha-Ostermann, Elisabeth Kamper, Janine Rippka, Cornelia Schwarz, Katharina Ehrenhöfer-Wölfer, Andreas Schlattl, Andreas Wernitznig, Jesse J Lipp, Gerd Bader, Ralph A Neumüller, Norbert Kraut, Mark A Pearson, Mark Petronczki, Simon Wöhrle: Full-time employee of Boehringer Ingelheim RCV GmbH & Co KG, Vienna, Austria. The other authors declare that no competing interests exist.

## Funding

| Funder | Grant reference number | Author |
|---|---|---|
| European Molecular Biology Organization | | Kota Nagasaka |
| Human Frontier Science Program | | Kota Nagasaka |
| Austrian Research Promotion Agency | FFG 834223 | Jan-Michael Peters |
| Austrian Research Promotion Agency | FFG 852936 | Jan-Michael Peters |
| H2020 European Research Council | 693949 | Jan-Michael Peters |

The funders had no role in study design, data collection and interpretation, or the decision to submit the work for publication.

## Author contributions

Simone Lieb, Silvia Blaha-Ostermann, Validation, Investigation, Visualization, Methodology, Writing—original draft; Elisabeth Kamper, Janine Rippka, Validation, Investigation, Visualization, Methodology, Writing—review and editing; Cornelia Schwarz, Validation, Investigation, Visualization, Methodology; Katharina Ehrenhöfer-Wölfer, Validation, Investigation, Visualization, Writing—review and editing; Andreas Schlattl, Andreas Wernitznig, Software, Writing—review and editing; Jesse J Lipp, Software, Formal analysis, Writing—review and editing; Kota Nagasaka, Software, Formal analysis, Validation, Investigation, Visualization, Writing—original draft; Petra van der Lelij, Validation, Investigation, Visualization, Writing—original draft; Gerd Bader, Formal analysis, Writing—review and editing; Minoru Koi, Ajay Goel, Conceptualization, Resources; Ralph A Neumüller, Conceptualization, Data curation, Software, Formal analysis, Validation, Visualization, Writing—original draft; Jan-Michael Peters, Resources, Supervision; Norbert Kraut, Mark A Pearson, Supervision, Funding acquisition, Project administration, Writing—review and editing; Mark Petronczki, Simon Wöhrle, Conceptualization, Supervision, Validation, Visualization, Methodology, Writing—original draft, Project administration

## Author ORCIDs

Kota Nagasaka http://orcid.org/0000-0003-0765-638X
Gerd Bader http://orcid.org/0000-0002-8410-3415
Jan-Michael Peters https://orcid.org/0000-0003-2820-3195
Mark Petronczki https://orcid.org/0000-0003-0139-5692
Simon Wöhrle http://orcid.org/0000-0002-2478-7333

## Decision letter and Author response

Decision letter https://doi.org/10.7554/eLife.43333.024
Author response https://doi.org/10.7554/eLife.43333.025

## Additional files

### Supplementary files

• Supplementary file 1. MSS/MSI-H status analysis of CRC, endometrial and gastric carcinoma cell lines. MSS/MSI-H status was analyzed using fluorescent PCR-based analysis of the mononucleotide microsatellite markers NR-21, BAT-26, BAT-25, NR-24 and MONO-27. Main peak sizes for the mononucleotide microsatellite markers are shown for the MSS control cell line K562 and CRC, endometrial and gastric carcinoma cell lines. Cell models were classified as MSS (blue) or MSI-H (red) according to the indicated size range classification of MSS alleles.
DOI: https://doi.org/10.7554/eLife.43333.016

• Supplementary file 2. Overview of cell lines used in this study. Cell lines used in this study are listed with tumor type of origin, MSS/MSI-H status, vendor source, and STR confirmation status. Variable STR profiles are reported for ISHIKAWA cells, consistent with MSI-H status (*Korch et al., 2012*).
DOI: https://doi.org/10.7554/eLife.43333.017

• Supplementary file 3. Sequences of sgRNAs used for CRISPR depletion studies. Sequences of sgRNAs used for targeting WRN are listed in N- to C-terminal order according to the representation in *Figure 3* and Expanded View *Figure 3*. Domains are annotated according to PFAM entry Q14191. RQC, RecQ helicase family DNA-binding domain; HRDC, Helicase and RNase D C-terminal, HTH, helix-turn-helix motif. Negative and positive control sgRNA sequences are also listed.
DOI: https://doi.org/10.7554/eLife.43333.018

• Transparent reporting form
DOI: https://doi.org/10.7554/eLife.43333.019

### Data availability

All data generated or analysed during this study are included in the manuscript and supporting files.

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
