## [Decision Letter]

Thank you for submitting your article "Werner syndrome helicase is a selective vulnerability of microsatellite instability-high tumor cells" for consideration by *eLife*. Your article has been reviewed by three peer reviewers, including Wolf-Dietrich Heyer as the Reviewing Editor and Reviewer #1, and the evaluation has been overseen by Jeffrey Settleman as the Senior Editor. The following individuals involved in review of your submission have agreed to reveal their identity: Ray Monnat (Reviewer #2); Petr Cejka (Reviewer #3).

The reviewers have discussed the reviews with one another and the Reviewing Editor has drafted this decision to help you prepare a revised submission.

Summary:

This intriguing new report identifies the WRN RecQ helicase as a new and potentially useful vulnerability in microsatellite-unstable ('MSI-high or MSI-H') cancer cell lines. The results are worth reporting after some additional work on the manuscript.

The authors identified a dependency of MMR-deficient cells on the RecQ-like helicase WRN, which constitutes a significant novel finding with far-reaching implications for improving treatment of MMR-deficient tumors. The selective dependency was identified in a large screen of 398 cancer cell models covering ~8,000 genes as published in 2017 and the correlation of MMR status and WRN dependency is highly significant (Figure 1). The observation is validated in multiple cell line models for MMR+ and MMR- colorectal cancer and endometrial cancer cell lines (Figure 2) and with WRN KO in 2 independent MMR+ and MMR- colorectal cancer cell lines (Figure 3) using viability and colony forming assays. Back-complementation activity depended on the WRN ATPase/helicase activity but not the nuclease activity in HCT 116 cells (Figure 4). The results with another MMR- cell line, RKO, are less clear (Figure 4—figure supplement 1, see below), suggesting the involvement of additional variables. Figure 5, Figure 6 and Figure 7 demonstrate that WRN knockdown in MMR-deficient cells leads to hallmarks of genomic instability including bridges and micronuclei (Figure 5), which is confirmed by life imaging (Figure 6) and endpoint analysis of mitotic spreads (Figure 7). The claims of the manuscript are largely well supported by the data with one exception (see essential points). Overall, the authors consistently use multiple cell lines and approaches to corroborate the dependence of MMR-deficient cells on WRN. The manuscript is very well written, and the results are displayed very clearly in the Figures and Tables.

Essential revisions:

1) Figure 4—figure supplement 1 and subsection “WRN dependency in MSI-H CRC is linked to its helicase function”: There is a significant difference between the back-complementation results in HCT 116 (Figure 4) and RKO (Figure 4—figure supplement 1) and the description in the text does not reflect the results of Figure 4—figure supplement 1. Figure 4 shows that the back-complementation with wildtype and nuclease-deficient WRN is very effective, whereas the ATPase/helicase-deficient mutant as well as the ATPase/helicase nuclease double mutant cannot complement the WRN siRNA knockdown cells. For RKO (Figure 4—figure supplement 1), the difference between nuclease and helicase mutants appears statistically insignificant as the error bars overlap. The text should be changed to more accurately reflect these results.

2) The authors meticulously show the dependence of the effect on WRN status, but the dependence on MMR status is only correlative. A key concern is that all experiments were carried upon depletion of WRN in MMR-deficient cell lines. These cell lines have a high mutator phenotype and accumulate mutations in a number of other genes, in particular in those containing microsatellites. Therefore, to confirm the key conclusion of this manuscript that the lethality is a result of simultaneous MMR and WRN deficiency, the experiment should also done the other way round, i.e. to deplete key MMR factors (MLH1 would be the best) in WRN-deficient cells, or to perform co-depletion experiments. Additionally, WRN depletion should also be carried out in HCT 116 +ch3 cell line, in which the expression of MLH1 was corrected.

HCT 116 and RKO are both MLH1 deficient. What is known about additional mutations in these cell lines that may contribute to this difference? This above approach might also address the difference between both cell lines. For RKO a complementation system was described in Yan et al., 2003.

3) WRN inhibitors need to be referenced and discussed more fully. In addition to the Rosenthal et al., 2010 reference, at least two WRN inhibitors have been around for 5+ years. They are highly and directly relevant to the topic of this manuscript, and are being pursued in other clinical contexts (see, e.g., Moles et al., 2016) that may provide additional insight. These inhibitors are readily available, and the authors' results make a straight-ahead prediction of what a dose-response outcome should be in their CRC cell line panels to provide a bit of pre-clinical data. The authors should acquire the WRN inhibitors and do dose-response curves on their MSI-H and MSS cell lines, and show that their WRN KO lines are insensitive (or largely insensitive) as a demonstration of target specificity.

[Editors' note: further revisions were requested prior to acceptance, as described below.]

Thank you for resubmitting your work entitled "Werner syndrome helicase is a selective vulnerability of microsatellite instability-high tumor cells" for further consideration at *eLife*. Your revised article has been favorably evaluated by Jeffrey Settleman (Senior Editor), a Reviewing Editor, and two reviewers.

The manuscript has been improved but there are some remaining issues that need to be addressed before acceptance, as outlined below:

The Abstract should be updated to reflect the significant insight that the MHL1 WRN synthetic lethality is context-dependent and not absolute.

*Reviewer #1:*

In this revision the authors meticulously addressed the comments made in the initial review. They conducted significant new experimentation that strengthens the conclusion about the role of the nuclease activity of WRN (essential revision 1).

The authors devoted significant efforts to address essential revision 2, conducting 3 lines of experimentation that clearly demonstrate that the MLH1 WRN synthetic lethality is not hard wired but is context dependent. This significantly affects the overall conclusion of the study, and it is very important that these new data were included in the revision and discussed. What I am missing is an update of the abstract to reflect this important point.

The authors also nicely addressed essential revision 3 testing 2 WRN inhibitors and 2 controls in their cell models. The results provide important insights about the inhibitors but do not contribute much mechanistic insight regarding the MLH1 WRN synthetic lethality. While the authors briefly refer to WRN inhibitors in the discussion, they elected not to include the data in the manuscript. I agree with their choice, as the inhibitors show little promise in terms of target specificity.

Point for final acceptance:

Update the Abstract to reflect the major conclusion that the MLH1 WRN synthetic lethality is context dependent and not absolute.

*Reviewer #3:*

The authors carried out experiments to address issues raised during the initial revision.

I now support publication of this paper. Although the effect seems "indirect" as far as MMR status is concerned, it is still very significant with regards to the therapeutic potential. Also, this is now a very competitive topic.

---

## [Author Response]

Essential revisions:1) Figure 4—figure supplement 1 and subsection “WRN dependency in MSI-H CRC is linked to its helicase function”: There is a significant difference between the back-complementation results in HCT 116 (Figure 4) and RKO (Figure 4—figure supplement 1) and the description in the text does not reflect the results of Figure 4—figure supplement 1. Figure 4 shows that the back-complementation with wildtype and nuclease-deficient WRN is very effective, whereas the ATPase/helicase-deficient mutant as well as the ATPase/helicase nuclease double mutant cannot complement the WRN siRNA knockdown cells. For RKO (Figure 4—figure supplement 1), the difference between nuclease and helicase mutants appears statistically insignificant as the error bars overlap. The text should be changed to more accurately reflect these results.

We agree with the reviewer’s analysis that in RKO cells the back-complementation studies do not clearly indicate the differential relevance of helicase and nuclease function in the context of WRN dependence. We have revised the text in the manuscript accordingly:

“In RKO cells expressing WRNr variants, we observed a slightly a stronger dependency on WRN helicase ATP-binding function compared to exonuclease activity upon knock-down of endogenous WRN”.

Since the determination of the relevant enzymatic function of WRN in MSI-H cancer cells is critical for both the mechanistic understanding and from a drug development perspective we have performed WRN back-complementation experiments in an additional WRN-dependent MSI-H cell model, the endometrial carcinoma cell line HEC-265. We generated monoclonal lines with stable expression of FLAG-tagged, siRNA-resistant WRN (WRNr) expression constructs harboring loss-of-function mutations within the exonuclease- (E84A, Nuclease-dead) and helicase-domain (K577M, ATP-binding deficient), or both domains (E84A/K577M, Double-mutant). Similar to the findings in HCT 116 cell we observed that exogenous expression of the nuclease-dead form of WRNr, but not the ATP-binding deficient and double-mutant forms rescued the effect of endogenous WRN depletion. WRNr expression levels were similar or higher for the mutant WRNr variants compared to WRNr wild-type.

This data, which we have included in the revised version of the manuscript as Figure 4—figure supplement 1C, is supportive for a predominant function of WRN helicase in MSI cancer cell lines.

2) The authors meticulously show the dependence of the effect on WRN status, but the dependence on MMR status is only correlative. A key concern is that all experiments were carried upon depletion of WRN in MMR-deficient cell lines. These cell lines have a high mutator phenotype and accumulate mutations in a number of other genes, in particular in those containing microsatellites. Therefore, to confirm the key conclusion of this manuscript that the lethality is a result of simultaneous MMR and WRN deficiency, the experiment should also done the other way round, i.e. to deplete key MMR factors (MLH1 would be the best) in WRN-deficient cells, or to perform co-depletion experiments. Additionally, WRN depletion should also be carried out in HC T116 +ch3 cell line, in which the expression of MLH1 was corrected.HCT 116 and RKO are both MLH1 deficient. What is known about additional mutations in these cell lines that may contribute to this difference? This above approach might also address the difference between both cell lines. For RKO a complementation system was described in Yan et al., 2003.

We agree that demonstrating whether WRN dependency of MSI-H cancer cell lines is attributable to an acute and context-independent synthetic lethal interaction or an acquired vulnerability related to the MMR-deficient status of these cell lines would substantially strengthen the relevance of this study. In line with the reviewers’ suggestions we have therefore addressed a potential co-dependence of WRN and MMR gene function with several approaches:

1) Test of differential WRN dependency in wild-type vs. MLH1-knock-out models of SW480 and CaCo-2.

2) Test of differential WRN dependency in parental vs. MLH1 and MLH1/MSH3 reconstituted HCT 116 cell lines (HCT 116 +ch3, HCT 116 +ch3/5).

3) Test of differential MLH1/MSH3 dependency in parental vs. WRN-knock-out models of SW480.

1) Test of differential WRN dependency in wild-type vs. MLH1-knock-out models of SW480 and CaCo-2.

To monitor whether loss of MLH1 would induce WRN dependency we generated MLH1-deficient clones in two MSS CRC cell lines (SW480 and CaCo-2) using CRISPR-Cas9 mediated gene knock-out and applied siRNA-mediated knock-down of WRN.

For both SW480- and CaCo-2-derived MLH1 knock-out clones we did not observe increased sensitivity to WRN knock-down compared to non-edited, wild-type clones. This indicates that at least in the context of established MSS CRC cancer cell lines MLH1 loss-of-function alone does not induce WRN dependency. Of note, WRN dependency was also not observed at later passages and MLH1 knock-out did not induce MSI in both models at >9 months cultivation post MLH1 knock-out.

**Author response image 1. respfig1:** SW480 (upper panel) and CaCo-2 (lower panel) parental and MLH1-knock-out monoclonal CRC cell lines were transfected with the indicated siRNAs. Cell viability was determined 7 days after transfection and is shown relative to non-targeting control (NTC) siRNA (n=3 biological replicates; error bars denote standard deviation). Knock-out of MLH1 was confirmed by immunoblotting.

2) Test of differential WRN dependency in parental vs. MLH1 and MLH1/MSH3 reconstituted HCT 116 cell lines (HCT 116 +ch3, HCT 116 +ch3 +ch5)

We thank the reviewer for pointing out the HCT 116 +ch3 cell line as a semi-isogenic, MLH1-proficient model to the parental MMR-deficient HCT 116 cell line used in our study. We have analyzed the effect of WRN knock-down in the HCT 116 +ch3 cells along with the MMR-deficient control line HCT 116 +ch2 and the HCT 116 +ch3 +ch5 line (with additional complementation of MSH3). We find that neither back-complementation of MLH1 alone or in combination with MSH3 reverses the strong WRN dependency of HCT 116 cells – although we observe a slightly increased viability upon WRN knock-down in the HCT 116 +ch3 and HCT 116 +ch3 +ch5 models compared to HCT 116 parental cells and the +ch2 reconstituted control model (Figure 2—figure supplement 2A). siRNA-mediated depletion of WRN mRNA was equally efficient in all cell models (Figure 2—figure supplement 2B).

3) Test of differential MLH1/ MSH3 dependency in parental vs. WRN-knock-out models of SW480.

To monitor whether loss of WRN would induce dependency on MLH1 and/or MSH3, we analyzed the effect of individual or combined knock-down of MLH1 and MSH3 on the viability of wild-type (WRN-proficient) and two WRN-knock-out monoclonal SW480 cell lines. Neither single nor concomitant knock-down of MHL1 and MSH3 affected cell viability of the WRN-deficient SW480 cell lines (Figure 2—figure supplement 3A) despite efficacious knock-down of both genes (Figure 2—figure supplement 3B). This data is in line with the studies outlined above and argue against an acute and direct synthetic lethal interaction of WRN and MLH1/MSH3 genes.

In summary, the results outlined above suggest that the WRN dependence of MSI-H/MMR-deficient cancer cell lines might not be attributable to a hard-wired, context independent synthetic lethality such as described for the paralog genes STAG1/STAG2 (van der Lelij et al., 2017) or SMARCA2/SMARCA4 (Oike et al., 2013). Instead, our data argue for an acquired dependency on WRN function as a consequence of persistent MSI-H status and/or MMR-deficiency.

To incorporate this relevant finding we have included the studies using the MLH1/*MLH3* reconstituted HCT 116 lines (2.) and the SW480 WRN-knock-out models (3.) in the revised version of this manuscript as Figure 2—figure supplement 2 and Figure 2—figure supplement 3. This is of particular relevance since our results are in partial discrepancy to a recent preprint manuscript reporting the full reversion of WRN dependency in the MLH1/MSH3 reconstituted HCT 116 +ch3 +ch5 model (Chan et al., 2019 bioRxiv preprint available at https://doi.org/10.1101/502070). In our experiments, we only detect a minor improvement in viability upon WRN depletion in HCT 116 +ch3 +ch5 model. We have also revised the Discussion section accordingly.

It is plausible that WRN dependency is a consequence of alterations in genetic targets of MSI (i.e. genes harboring repetitive DNA sequences that are particularly vulnerable to the loss of MMR activity). To this end, we analyzed the effect of siRNA-mediated knock-down of genes commonly mutated in MSI-H cancer cell lines on viability of parental vs. WRN-knock-out SW480 cell lines. We did not observe an increased dependency on any of the genes tested the context of concomitant WRN loss. This argues against an acute synthetic lethal interaction of WRN and MSI target genes. However, we cannot rule out synthetic lethal interactions based on combined loss of MSI target genes and have addresses this accordingly in the Discussion section of the revised manuscript.

**Author response image 2. respfig2:** Left panel: List of MSI target genes and positive/negative control genes included in the siRNA knock-down analysis. Cyclophilin was included as a negative control gene; PLK1, PSMA1 and SGO1 are used as positive control genes. siRNA targeting WRN was also include in the analysis. NTC, non-targeting control siRNA. Right panel: Effect of siRNA-mediated knock-down of MSI target gens on viability of parental and WRN-knock-out SW480 cell lines.

3) WRN inhibitors need to be referenced and discussed more fully. In addition to the Rosenthal et al., 2010 reference, at least two WRN inhibitors have been around for 5+ years. They are highly and directly relevant to the topic of this manuscript, and are being pursued in other clinical contexts (see, e.g., Moles et al., 2016) that may provide additional insight. These inhibitors are readily available, and the authors' results make a straight-ahead prediction of what a dose-response outcome should be in their CRC cell line panels to provide a bit of pre-clinical data. The authors should acquire the WRN inhibitors and do dose-response curves on their MSI-H and MSS cell lines, and show that their WRN KO lines are insensitive (or largely insensitive) as a demonstration of target specificity.

We thank the reviewers for pointing out the use of published WRN inhibitors in the context of this study and we agree that it is highly relevant to assess the potential clinical value of our findings with the available chemical matter.

We have tested two structurally related WRN inhibitors, NSC 19630 (Aggarwal et al., 2011) and NSC 617145 (Aggarwal et al., 2013) and a chemical probe for the RecQ helicase BLM (Rosenthal et al., 2010) across a panel of MSS and MSI cell models. Both NSC 19630 and NSC 617145 affected cell viability in 7-day dose-response proliferation assays with IC_50_ values below 10 µM in the majority of the tested cell lines (see Author response table 1). However, MSI-H CRC and endometrial carcinoma cell lines did not show increased sensitivity to treatment with the two WRN inhibitors compared to MSS cell models. The BLM inhibitor ML-216 was largely ineffective in the majority of cell lines at concentrations up to 10 µM. As a positive control, we included the pan-HDAC inhibitor panobinostat which showed potent inhibition of cell viability in the nM range across all cell models.

We have included two SW480 monoclonal cell lines with an engineered deletion of WRN in order to assess the on-target activity of the WRN inhibitors. Both SW480 WRN knock-out lines display sensitivities lower or similar to the parental cell line or a WRN-proficient SW480 monoclonal cell line. In the absence of the protein target this indicates that the observed toxicity of the WRN inhibitors might be at least in part attributable to non-WRN related off-target effects.

While these results do not preclude on-target activities of the tested WRN inhibitors in line with the literature data, we suspect that the inhibitors lack the potency and selectivity required for the cellular assessment of WRN function in the context of this study. Since the potential off-target activities pose a challenge in interpreting the cellular effects compared to the genetic inactivation of WRN, we have not included this data in the revised version of this manuscript. We have included the references to the WRN inhibitors NSC 19630 and NSC 23867477 in the revised Discussion section of the manuscript.

**Author response table 1. resptable1:** Cell lines were plated at initial densities of 1000 or 2000 cells per well of a 96-well plate. The following day, cells were treated with the indicated inhibitors at concentrations ranging from 10 to 0.04 µM (0.5 to 0.002 µM for panobinostat). Cell viability was determined 7 days after treatment. Data for NSC 19630, ML-216 and panobinostat are representative IC_50_ values from three experiments, NSC 617145 IC_50_ values are from a single analysis.

**Tumor Type**	**MSS/MSI status**	**Cell line**	**NSC 617145** **IC_50_ [µM]**	**NSC 19630** **IC_50_ [µM]**	**ML-216** **IC_50_ [µM]**	**Panobinostat** **IC_50_ [µM]**
CRC	MSS	SK-CO-1	7.09	3.11	>10	0.011
MSS	SW480	2.14	3.32	>10	0.028
MSS	SW480_wt_clone	1.83	4.09	>10	0.018
MSS	SW480_WRN_KO_clone#1	1.73	2.64	>10	0.016
MSS	SW480_WRN_KO_clone#2	1.38	3.73	>10	0.022
MSI	HCT 116	8.87	4.99	4.40	0.012
MSI	RKO	>10	6.39	>10	0.044
Endometrial carcinoma	MSS	MFE-280	4.54	3.38	5.71	0.009
MSI	HEC-265	7.74	6.80	>10	0.021
MSI	ISHIKAWA	>10	5.55	>10	0.016
Non-transformed	MSS	hTERT-RPE-1	4.34	5.54	>10	0.035

[Editors' note: further revisions were requested prior to acceptance, as described below.].

Reviewer #1:[…]Point for final acceptance:Update the Abstract to reflect the major conclusion that the MLH1 WRN synthetic lethality is context dependent and not absolute.

The Abstract has been updated to reflect the new experimentation. We have included a statement that WRN dependence is not acutely linked to loss of MMR gene function, but rather a consequence of sustained MMR-deficiency. The Abstract has further been slightly modified to adhere to the 150 word limitation despite addition of the new information.